# HaSa: Hardness and Structure-Aware Contrastive Knowledge Graph Embedding

## ABSTRACT

We consider a contrastive learning approach to knowledge graph embedding (KGE) via InfoNCE. For KGE, efficient learning relies on augmenting the training data with negative triples. However, most KGE works overlook the bias from generating the negative triples—false negative triples (factual triples missing from the knowledge graph). We argue that the generation of high-quality (i.e., hard) negative triples might lead to an increase in false negative triples. To mitigate the impact of false negative triples during the generation of hard negative triples, we propose the Hardness and Structure-aware (**HaSa**) contrastive KGE method, which alleviate the effect of false negative triples while generating the hard negative triples. Experiments show that HaSa improves the performance of InfoNCE-based KGE approaches and achieves state-of-the-art results in several metrics for WN18RR datasets and competitive results for FB15k-237 datasets compared to both classic and pre-trained LM-based KGE methods.

## CCS CONCEPTS

• **Computing methodologies** → **Knowledge representation and reasoning**; **Feature selection**.

## KEYWORDS

Knowledge Graph Embedding, Contrastive Learning, Negative Sampling

**ACM Reference Format:**

Anonymous Author(s). 2024. HaSa: Hardness and Structure-Aware Contrastive Knowledge Graph Embedding. In *Proceedings of Make sure to enter the correct conference title from your rights confirmation email (Conference acronym 'XX)*. ACM, New York, NY, USA, 10 pages. https://doi.org/XXXXXXX.XXXXXXX

## 1 INTRODUCTION

A knowledge graph (KG) is a structured representation of factual descriptions consisting of entities and relationships. Formally, a knowledge graph is defined as a triple database $\mathcal{G} = (\mathcal{E}, \mathcal{R}, \mathcal{T})$ where $\mathcal{E}$ and $\mathcal{R}$ are the entity set and relationship set, and $\mathcal{T} = \{(h, r, t) | h, t \in \mathcal{E}, r \in \mathcal{R}\}$ is the triple set, where $h$ is the head entity, $t$ is the tail entity and $r$ is the relationship. Each triple $(h, r, t)$ captures a fact, for example (**Obama, born in, Honolulu**).

Knowledge graph embedding (KGE), also known as knowledge representation learning (KRL), aims to learn a deterministic embedding function $f(\cdot)$ that maps the $h$, $r$, and $t$ to a lower dimensional space $f(h) = \mathbf{e}_h \in \mathbb{R}^d$, $f(r) = \mathbf{e}_r \in \mathbb{R}^d$, and $f(t) = \mathbf{e}_t \in \mathbb{R}^d$ [14, 33]; some KGE methods may map $r$ to a separate space from the entities. KGE can be used in many applications such as question-answering [19, 21] and drug discovery [17]. An important downstream application is knowledge graph completion (also called link prediction problem): given a query, $hr$, by combining a head entity $h$ and relationship $r$, the goal is to infer the corresponding tail entity $t$ from a set of candidate entities. For example, given the query (**Obama, born in**), link prediction algorithms are trained to correctly predict the corresponding tail entity **Honolulu**.

Recently, contrastive learning, a self-supervised learning method, has shown good performance for embedding problems in computer vision [5, 12] and natural language processing [9], knowledge graph embedding [27, 31, 32]. Contrastive learning augments the training data by creating new data samples. How new data are generated depends on what kind of data we want to embed. In KGE, a given *positive triple* from the knowledge graph, $(h, r, t) \in \mathcal{T}$, can be augmented by replacing the original tail $t$ with another tail entity $t^-$ such that the new triple $(h, r, t^-) \notin \mathcal{T}$. This creates a *negative triple*. For each positive triple in $\mathcal{T}$, we need to generate multiple negative triples. InfoNCE[22] is a popular loss function in contrastive learning.

Exhaustively generating all possible negative triples for training is expensive (i.e., replace $t$ by all other entities in $\mathcal{E}$). Instead, we only generate $K$ negative tails. Simple InfoNCE[22] samples negative tails from a distribution that is independent of the query. While this approach is simple to implement, it was shown that higher-quality negative samples, called *self-adversarial* or *hard* negative samples, lead to better performance [13, 15]. Hard InfoNCE samples negative tails from a distribution that depends (semantically and lexicographically) on the query embedding. For example, (**New York, location adjoining, New Caledonia**) and (**New York, location adjoining, Avatar Movie**) are both negative triples in a knowledge graph. However, (**New York, location adjoining, New Caledonia**) is more useful than (**New York, location adjoining, Avatar Movie**) for learning a meaningful embedding. Thus, we expect that the former triple would be harder than the latter one with respect to the majority of standard metrics used in natural language processing.

Empirical studies show that hard negative triples need to be generated carefully since they tend to be *false negative triples*, which are negative triples that are factual. False negative triples should be in $\mathcal{T}$ but are not due to the incompleteness of the knowledge graph. False negative triples will cause the augmented training set to be biased and have a negative effect on learning the embedding function.

In this paper, we modify InfoNCE loss for KGE to alleviate the effect of false negative triples while still keeping the advantage of hard negatives. For a particular query $(h, t)$, we generate a hard negative tail, $t^-$, using the same sampling distribution as Hard InfoNCE (this accounts for the query embedding). We weigh the resultant hard negative triple, $(h, r, t^-)$, by the probability that it is *not* a false negative triple. We utilize the shortest path length between $h$ and $t^-$ in the knowledge graph structure to approximate this probability. We call our method Hardness and Structure-aware (HaSa) contrastive KGE. Furthermore, we also boost the HaSa by considering the bi-directional loss (HaSa+). Our experiments using Wn18RR and FB15k-237 datasets show that HaSa is better than the InfoNCE-based method without considering false negative triples. More broadly, HaSa and HaSa+ methods perform on par with the wide range of comparable state-of-the-art solutions for KGE. In summary, our contributions are as follows:

(1) Experimentally using WN18RR and FB15k-237, Hard InfoNCE, which samples, $t^-$, from a distribution that depends on the query embedding, generates much more false negative triples than Simple InfoNCE. We discover that hard negative triples with smaller shortest path lengths (between $h$ and $t^-$ in the knowledge graph structure) are more likely to be false negative triples.

(2) We propose HaSa, which weighs hard negative triples by the probability that they are false negative triples. This reduces the effect of false negative triples on the loss calculation. Experiments show that HaSa has enhanced link prediction results compared to Simple InfoNCE and Hard InfoNCE.

(3) We performed our method HaSa and HaSa+ methods on the WN18RR and FB15k-237 datasets. The experimental results on par in link prediction tasks with comparable state-of-the-art KGE methods on both classic methods such as RotatE [26], ComplexE [29] and pre-trained language-based methods such as StAR [30], LASS [25].

## 2 RELATED WORK

**Knowledge graph embedding** is often used for knowledge graph completion. KGE methods need to be trained on the negative triples to differentiate from the original positive triples. Early works such as Bordes et al. [3], Lin et al. [18], Yang et al. [36] used triplet loss, by assigning higher scores to positive triples than negative triples. Trouillon et al. [29], Xu and Li [35] used negative log-likelihood while Dettmers et al. [8], Yao et al. [38] used cross-entropy loss. More recent works such as Shen et al. [25], Sun et al. [26] used negative sampling loss proposed by [20], which is similar to noise contrastive estimation (NCE) [11]. More recently, InfoNCE loss [22] has shown significant improvements in contrastive learning, leading to the proposal of several KGE methods based on InfoNCE loss[27, 31, 32, 37].

Apart from such model-based loss functions, Some KGE methods use various heuristics to investigate the loss function by considering the data-driven loss function to generate useful negative triples for efficient training. For instance, Cai and Wang [4], Sun et al. [26] assigned a probability to each negative triple based on its plausibility in the current embedding space. Yao et al. [38] leveraged textual information from the knowledge graph to propose KG-BERT. Shen et al. [25], Wang et al. [30] combined structural information with pre-trained language models (LMs). Recently, Wang et al. [32] proposed concatenating the head entity and relationship and feeding them into a pre-trained model. In addition to the embedding space, Ahrabian et al. [1], Wang et al. [30] explored building negative triples based on graph topology. Wang et al. [31] introduced a temperature constant in the contrastive loss to control the hardness of negative triples. However, previous research did not analyze the effectiveness of hard negatives and neglected the issue of false negative triples.

Our work is also related to the role of negatives in **contrastive learning**, a self-supervised learning method that has shown impressive results in many different applications. The quality of negative samples plays a crucial role in the effectiveness of contrastive learning. Previous studies [15, 24, 34] have demonstrated that using hard negatives can enhance contrastive learning results. Other approaches [7, 16] have modified the InfoNCE loss to address the issue of false negative samples in image classification. More recently, some methods [6, 10] replaced negative samples by using momentum update or stop-gradient.

## 3 BACKGROUND: SIMPLE INFONCE LOSS

InfoNCE loss [22] was used to learn optimal embedding for audio, images, and natural language tasks. It has since been adopted for KGE [27, 31].

In the KG, for a given triple $(h, r, t) \in \mathcal{T}$, we will sample $K$ independent and identically distributed negative tails, $t_j^-, j = 1, \ldots K$ from the negative sample distribution $p^-(t)$ which independent to the $h$ and $t$. This will make $K$ negative triples: $(h, r, t_j^-)$. Let $f(t_j^-) = \mathbf{e}_{t_j}^-$ denote the embedding of the negative tail $t_j^-$.

The InfoNCE loss adopted to KGE is

$$\mathcal{L}_{Info} = \sum_{(h,r,t) \in \mathcal{T}} \left[ -\log \left( \frac{\exp(\mathbf{e}_{hr}^T \mathbf{e}_t)}{\exp(\mathbf{e}_{hr}^T \mathbf{e}_t) + \sum_{j=1}^K \exp(\mathbf{e}_{hr}^T \mathbf{e}_t^-)} \right) \right], \quad (1)$$

where $\mathbf{e}_{hr}$ is the query embedding obtained with an aggregation function $g(\cdot, \cdot) : \mathbb{R}^d \times \mathbb{R}^d \to \mathbb{R}^d$ such that $g(f(h), f(r)) = g(\mathbf{e}_h, \mathbf{e}_r) = \mathbf{e}_{hr} \in \mathbb{R}^d$. We follow the setting of InfoNCE[22] to use the gated recurrent unit (GRU) neural network to model the aggregation function $g(\cdot, \cdot)$.

By minimizing $\mathcal{L}_{Info}$, we aim to learn $f(\cdot)$ and $g(\cdot)$ such that the query and the corresponding positive tail are mapped to vectors (in the embedding space) that are close together, while the query and the negative tails are mapped to vectors that are far apart.

Assuming that the joint distribution of $\mathbf{e}_{hr}, \mathbf{e}_t, \mathbf{e}_{t_1}^-, \ldots \mathbf{e}_{t_K}^-$ can be factored as $p(\mathbf{e}_{hr}, \mathbf{e}_t) \prod_{j=1}^K p^-(\mathbf{e}_{t_j}^-)$, [22] showed that the InfoNCE loss gives a lower bound on the mutual information between $\mathbf{e}_{hr}, \mathbf{e}_t$. That is, by minimizing InfoNCE loss in KGE, we are maximizing the mutual information between query and corresponding tail embedding, $I(\mathbf{e}_{hr}; \mathbf{e}_t)$. Aitchison [2] also showed that the infoNCE objective is equal (up to a constant) to the log Bayesian model evidence.

To explicitly account for the negative sample distribution, we can formulate the InfoNCE loss as

$$\mathcal{L}_{Simple\_Info} =$$
$$\sum_{(h,r,t)\in\mathcal{T}}\left[-\log\left(\frac{\exp(\mathbf{e}_{hr}^T\mathbf{e}_t)}{\exp(\mathbf{e}_{hr}^T\mathbf{e}_t)+K\cdot\mathbb{E}_{t^-\sim p^-(t)}[\exp(\mathbf{e}_{hr}^T\mathbf{e}_t^-)]}\right)\right], \quad (2)$$

where $p^-(t)$ is the negative sample distribution. We can assume that

$$p^-(t) = \frac{\#(t)}{\sum_{t'\in\mathcal{E}_{batch}}\#(t')}, \qquad (3)$$

where $\mathcal{E}_{batch}\subseteq\mathcal{E}$ is the subset consisting of entities in the training batch and $\#(t)$ is the number of times $t$ occurs as a tail entity in training triple batch $\mathcal{T}_{batch}\subseteq\mathcal{T}$. The negative tails sampled from $p^-(t)$ are considered simple because they are generated independently from the query.

To see the relationship between (1) and the original InfoNCE loss (2) proposed in [22], it is enough to consider $K$ samples of negative tails $\{t_j^-\}$ from $p^-(t)$ and the approximation

$$\mathbb{E}_{t^-\sim p^-(t)}[\exp(\mathbf{e}_{hr}^T\mathbf{e}_t^-)] \approx \frac{1}{K}\sum_{j=1}^{K}\exp(\mathbf{e}_{hr}^T\mathbf{e}_t^-).$$

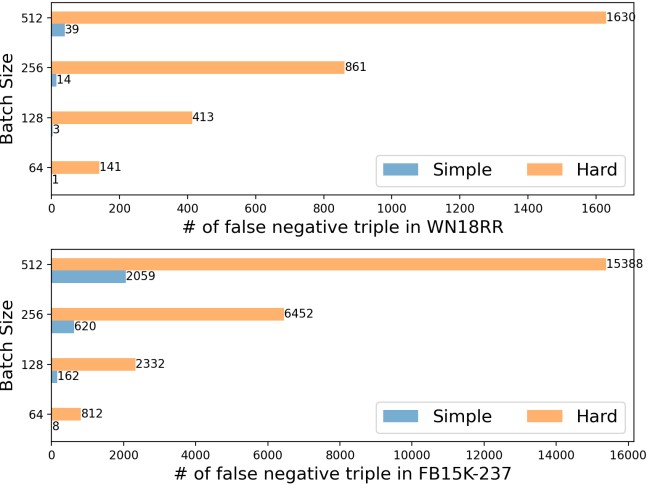

**Figure 1: Number of false negative triples generated using Simple $p^-(t)$ and Hard $p^-(t|\mathbf{e}_{hr})$ methods.**

# 4 INFONCE LOSS WITH HARD NEGATIVE TRIPLES

In practice, KGE algorithms often use heuristics to generate hard negative triples to maximize performance. Hard negative triples are *harder* to distinguish from the triples in the KG than arbitrarily generated negative samples[13, 15]. One way to generate hard negative triples is to sample the tail entity from a negative sample distribution that also considers the query phrase. That is, we sample the negative tail from the negative sample distribution $p^-(t|\mathbf{e}_{hr})$.

InfoNCE loss function with hard negative triples becomes

$$\mathcal{L}_{Hard\_Info} =$$
$$\sum_{(h,r,t)\in\mathcal{T}}\left[-\log\left(\frac{\exp(\mathbf{e}_{hr}^T\mathbf{e}_t)}{\exp(\mathbf{e}_{hr}^T\mathbf{e}_t)+K\mathbb{E}_{t^-\sim p^-(t|\mathbf{e}_{hr})}[\exp(\mathbf{e}_{hr}^T\mathbf{e}_t^-)]}\right)\right]. \quad (4)$$

Similar to RotatE [26] and KBGAN [4], we use the following negative sample distribution:

$$p^-(t|\mathbf{e}_{hr}) = \frac{\exp(\mathbf{e}_{hr}^T\mathbf{e}_t)}{\sum_{t'\in\mathcal{E}_{batch}}\exp(\mathbf{e}_{hr}^T\mathbf{e}_{t'})}, \qquad (5)$$

which gives preference to tail entities whose embedding is close to the query embedding.

At the beginning of the learning process, we do not know the embedding $(f(\cdot))$ and aggregation $(g(\cdot))$ functions. Therefore, we initialize $\mathbf{e}_{hr}, \mathbf{e}_t, \mathbf{e}_t^-$ using representations from a pre-trained LMs.

## 4.1 Hard Negative Triples may be False Negative Triples

Reference [7] discussed the possibility of *false negative* samples, which are negative samples that (inadvertently) share the same class label as the original data. It was argued that they would hurt the downstream task.

In KGE, false negative triples will hurt embedding. The false negative tail embedding should be close to the query embedding (since the triple is factually true) but it's instead, pulled away from the query embedding by the gradient update during the training process (See the rigorous analysis in **Remark 1** A.1).

First, we will see that using (5) to generate negative tails results in more false negative triples than using (3). We considered two benchmark knowledge graph datasets: WN18RR[8] and FB15k-237[28]. We randomly removed 30% of the triples in the training. We call the set of removed triples $\mathcal{T}_{missing}$ and the set of triples that we retain $\mathcal{T}_{retain}$.

For every triple $(h,r,t)$ in $\mathcal{T}_{retain}$, we will generate $K$ negative triples by sampling the negative tail, $t^-$, from 1) $p^-(t)$, as defined in (3), 2) $p^-(t|\mathbf{e}_{hr})$, as defined in (5). If the negative triple, $(h,r,t^-)$, can be found in $\mathcal{T}_{missing}$, then it is factual and is a *false negative triple*. Otherwise, it is considered a *true negative triple*. This is a weak assumption since we have no way to know if this is correct.

Note that the number of negative triples, $K$, depends on the batch size. For each triple $(h,r,t)\in\mathcal{T}_{batch}$, we generate $K = 2(|\mathcal{T}_{batch}|) - 1$ negative triples.

Figure 1 shows the total number of false negative triples we obtained for WN18RR and FB15K-237 when we consider the different numbers of $K$. We can see that the hard negative sample distribution $p^-(t|\mathbf{e}_{hr})$ produces far more false negatives than the simple negative sample distribution $p^-(t)$.

## 4.2 Shortest Path Length Distinguishes True and False Negative Triples

Unless we consult an external source, we do not know with certainty if a negative triple $(h,r,t^-)$ is factual (true negative tripe) or not (false negative triple). However, we found that we can (approximately) differentiate between true and false negative triples using

the knowledge graph structure. Several KGE methods have utilized the structural information of $\mathcal{G}$ [1, 30].

Consider an unweighted, undirected graph $\mathcal{G}$ induced by the triples in the knowledge graph. The nodes of $\mathcal{G}$ represent the entities, and the edges represent relations. Let $d(h, t)$ denote the shortest path length from a node representing the head entity, $h$, to another node representing the tail entity, $t$, in $\mathcal{G}$.

Figure 2 shows the histogram of $d(h, t)$ for true and false negative triples generated by hard negative sampling from $p^-(t|\mathbf{e}_{hr})$. We can see that in both WN18RR and FB15k-237, the false negative triples tend to have smaller $d(h, t)$ than true negative triples. We will leverage this observation to mitigate the impact of false negative triples.

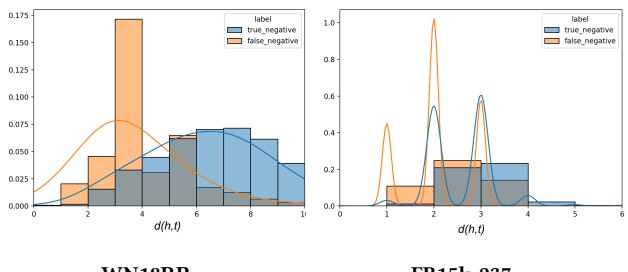

**WN18RR**                **FB15k-237**

**Figure 2: Histogram of shortest path length $d(h, t)$ between head entity $t$ and tail entity $t$ for true and false hard negative triples**

# 5 HARDNESS AND STRUCTURE-AWARE (HASA) CONTRASTIVE KGE

As we showed in the previous section, hard negative triples may be false negative triples, which degrades embedding performance. Chuang et al. [7] proposed the debiased contrastive loss, which accounts for false negative samples.

We can similarly modify the InfoNCE loss with hard negatives by considering a latent variable $\ell$ labeling triples as factual or non-factual: $\ell \in \{\text{fact}, \text{nonfact}\}$. The ideal hard negative sample distribution should be $p^-(t|\mathbf{e}_{hr}, \ell = \text{nonfact})$ instead of $p^-(t|\mathbf{e}_{hr})$.

We use the graph structure information of the knowledge graph to filter the false negative triples out of the hard negative triples obtained with (5). We call it hardness and structure-aware (HaSa) loss function

$$\mathcal{L}_{HaSa} =$$
$$\sum_{(h,r,t)\mathcal{T}} -\log\left(\frac{\exp(\mathbf{e}_{hr}^T \mathbf{e}_t)}{\exp(\mathbf{e}_{hr}^T \mathbf{e}_t) + K\mathbb{E}_{t^-\sim p^-(t|\mathbf{e}_{hr}, \ell=\text{nonfact})}[\exp(\mathbf{e}_{hr}^T \mathbf{e}_t^-)]}\right). \tag{6}$$

Since we can not know if a negative triple is factual, we can not directly sample the negative sample distribution as before. Using the Law of Total Expectation, we see that

$$\mathbb{E}_{t^-\sim p^-(t|\mathbf{e}_{hr}, \ell=\text{nonfact})}[\exp(\mathbf{e}_{hr}^T \mathbf{e}_t^-)] =$$
$$\frac{1}{1-\tau}\mathbb{E}_{t^-\sim p^-(t|\mathbf{e}_{hr})}[\exp(\mathbf{e}_{hr}^T \mathbf{e}_t^-)] - \frac{\tau}{1-\tau}\mathbb{E}_{t^-\sim p^-(t|\mathbf{e}_{hr}, \ell=\text{fact})}[\exp(\mathbf{e}_{hr}^T \mathbf{e}_t^-)], \tag{7}$$

where $\tau = p(\ell = \text{fact}|\mathbf{e}_{hr})$ and $1 - \tau = p(\ell = \text{nonfact}|\mathbf{e}_{hr})$ are hyperparameters that we will learn set via ablation study (see Section 7.4). The distribution of false negative tails $p^-(t|\mathbf{e}_{hr}, \ell = \text{fact})$ is remarkable and concentrate on the $d(h, t)$ region. Approximating the $p^-(t|\mathbf{e}_{hr}, \ell = \text{fact})$ is easier and it has much smaller sample space than $p^-(t|\mathbf{e}_{hr}, \ell = \text{nonfact})$. Thus, instead of estimating the expectation in terms of the true negative triples, we estimate the expectation in terms of the false negative triples.

Note that we can approximate the expectation in the first term on the right-hand side of (7) based on the hard negative sample distribution (5). The second term of equation 7 requires samples from $p^-(t|\mathbf{e}_{hr}, \ell = \text{fact})$, which we approximate with

$$p^-(t|\mathbf{e}_{hr}, \ell = \text{fact}) \propto \exp(\mathbf{e}_{hr}^T \mathbf{e}_t)\alpha(t|\mathbf{e}_{hr}), \tag{8}$$

where $\alpha(t|\mathbf{e}_{hr})$ depend on the shortest path $d(h, r)$ in $\mathcal{G}$.

Let $\mathcal{N}_1(h)$ be the set of nodes in $\mathcal{G}$ whose shortest path length from a head node $h$ is one. By definition, these nodes would be the set of tail nodes of $h$. Let $\mathcal{N}_2(h)$ be the set of nodes whose shortest path length from a head node $h$ is two, these nodes may be heads or tails nodes. We construct $\alpha$ to be

$$\alpha(t|\mathbf{e}_{hr}) = \begin{cases} \frac{1}{|\mathcal{N}_1(h)|+|\mathcal{N}_2(h)|}, & \text{if } d(h, t) \leq 2 \\ 0, & \text{otherwise.} \end{cases} \tag{9}$$

As we showed in Section 4.2, the shortest path length can distinguish between true and false negative triples. The second expectation can be approximated via importance sampling.

$$\mathbb{E}_{t^-\sim p^-(t|\mathbf{e}_{hr}, \ell=\text{fact})}[\exp(\mathbf{e}_{hr}^T \mathbf{e}_t^-)] \tag{10}$$
$$= \mathbb{E}_{t^-\sim\alpha(t|\mathbf{e}_{hr})}\left[\exp(\mathbf{e}_{hr}^T \mathbf{e}_t^-)\frac{p^-(t|\mathbf{e}_{hr}, \ell = \text{fact})}{\alpha(t|\mathbf{e}_{hr})}\right] \tag{11}$$
$$/ \mathbb{E}_{s^-\sim\alpha(t|\mathbf{e}_{hr})}\left[\frac{p^-(t|\mathbf{e}_{hr}, \ell = \text{fact})}{\alpha(t|\mathbf{e}_{hr})}\right] \tag{12}$$
$$\approx \sum_{m=1}^M \frac{\exp(2\mathbf{e}_{hr}^T \mathbf{e}_{t_m}^-)}{\sum_{m=1}^M \exp(\mathbf{e}_{hr}^T \mathbf{e}_{t_m}^-)} \tag{13}$$

where $\mathbf{e}_{t_m}^-$ are the embedding of the $M$ Monte Carlo samples, $s_m$ from $\alpha(t|\mathbf{e}_{hr})$.

With $K$ samples $\{t_j^-\}$ from (5) and $M$ samples $\{s_m^-\}$ from $\alpha(t|\mathbf{e}_{hr})$,

$$\mathbb{E}_{t_j^-\sim p^-(t|\mathbf{e}_{hr}, \ell=\text{nonfact})}[\exp(\mathbf{e}_{hr}^T \mathbf{e}_t^-)] \approx$$
$$\frac{1}{1-\tau}\sum_{j=1}^K \frac{\exp(2\mathbf{e}_{hr}^T f(t_j^-))}{\sum_{j=1}^K \exp(\mathbf{e}_{hr}^T f(t_j^-))} - \frac{\tau}{1-\tau}\sum_{m=1}^M \frac{\exp(2\mathbf{e}_{hr}^T f(s_m^-))}{\sum_{m=1}^M \exp(\mathbf{e}_{hr}^T f(s_m^-))}. \tag{14}$$

The formal steps of HaSa are presented as Algorithm 1.

# 6 IMPROVED HASA: HASA+

We can make one additional modification to the HaSa loss to improve performance. Thus far, for a given query (head & relationship) from triple $(h, r, t) \in \mathcal{T}$, we have discussed different methods to generate $K$ negative tails $t^-$, so that the generated negative triples $(h, r, t^-)$ are useful for learning a good knowledge graph embedding. Similarly, we can consider keeping the tail entity, $t$, and generating $K$ negative contexts, $h^-, r^-$, so the the negative triples $(h^-, r^-, t)$

---

**Algorithm 1:** An algorithm of HaSa

**Input** : Batch of triple $\mathcal{T}_{batch}$, $\tau$, graph structure $\mathcal{G}$, current encoder $f$.

**Output**: The loss $\mathcal{L}_{HaSa}$.

1 Extract entity batch $\mathcal{E}_{batch}$ from $\mathcal{T}_{batch}$;

   **for** $(h, r, t)$ *in* $\mathcal{T}_{batch}$ **do**

2      |  $\mathbf{e}_h = f(h)$;

      |  $\mathbf{e}_r = f(r)$;

      |  $\mathbf{e}_t = f(t)$;

      |  $\mathbf{e}_{ht} = g(\mathbf{e}_h, \mathbf{e}_r)$;

      |  $\mathcal{E}_{hard} = HardNegative\,(\mathbf{e}_{ht}, \mathcal{E}, f)$;

      |  $\{t_j^-\} = \mathcal{E}_{batch} \cup \mathcal{E}_{hard}/\{t\}$ ;

      |  $\{s_i^-\}$ = Sampling neighbour nodes based on the

      |    distribution $\alpha(\cdot | \mathbf{e}_{ht})$ (equation 9);

      |  Pos = $\exp(\mathbf{e}_{ht}^T \mathbf{e}_t)$;

      |  Neg = $\frac{1}{|K|} \sum_{t_j^-} \exp(\mathbf{e}_{ht}^T \mathbf{e}_{t_j}^-)$;

      |  FalseNeg = $\frac{1}{|M|} \sum_{s_i^-} \exp\,(\mathbf{e}_{ht}^T \mathbf{e}_{s_i}^-)$ ;

      |  NegHasa = $K(\frac{1}{(1-\tau)} \text{Neg} - \tau \text{FalseNeg})$ ;

      |  Calculate $\mathcal{L}_{HaSa}(h, r, t)$ for each triple

      |    $\mathcal{L}_{HaSa}(h, r, t) = \text{Pos}/(\text{Pos} + \text{NegHasa})$;

   **end**

3 $\mathcal{L}_{HaSa} = \sum_{(h,r,t) \in \mathcal{T}_{batch}} \mathcal{L}_{HaSa}(h, r, t)$;

---

**Algorithm 2:** *HardNegative*

**Input** : Entity set $\mathcal{E}$, $\mathbf{e}_{ht}$, current encoder $f$

**Output**: $\mathcal{E}_{hard}$

1 Filter the $\mathcal{E}$ to get rid of positive tails corresponding $h$ based on training dataset;

2 **for** $e$ in $\mathcal{E}$ **do**

   |  $\phi(\mathbf{e}_{ht}, e) = \mathbf{e}_{ht}^T f(e)$

   **end**

3 Sorted the $e$ based on the value of $\phi(\mathbf{e}_{ht}, e)$ and extract $\mathcal{E}_{hard}$;

---

can be also be used. This sort of bi-directional contrasting has been done in computer vision literature [5].

A disadvantage of considering negative contexts is that the support of the negative sample distribution $p^-(h, r)$ is $|\mathcal{E}| \times |\mathcal{R}|$, which can be prohibitively large. Therefore, we only consider simple negative sample distribution for the context.

The final modification to the InfoNCE loss is

$$\mathcal{L}_{HaSa+} =$$
$$\sum_{(h,r,t) \in \mathcal{T}} -\log \left( \frac{\exp(\mathbf{e}_{hr}^T \mathbf{e}_t)}{\exp(\mathbf{e}_{hr}^T \mathbf{e}_t) + \sum_{j=1}^{K} \mathbb{E}_{t_j^- \sim p^-(t|\mathbf{e}_{hr}, \ell=\text{nonfact})} \left[ \exp(\mathbf{e}_{hr}^T \mathbf{e}_t^-) \right]} \right)$$
$$- \log \left( \frac{\exp(\mathbf{e}_{hr}^T \mathbf{e}_t)}{\exp(\mathbf{e}_{hr}^T \mathbf{e}_t) + \sum_{j=1}^{K} \mathbb{E}_{h^-, r^- \sim p^-(h,r)} \left[ \exp(\mathbf{e}_t^T \mathbf{e}_{(hr)_j}^-) \right]} \right).$$
$$(15)$$

The algorithm for HaSa+ can be found in Appendix A.2.

# 7 EXPERIMENTS

**Dataset:** We considered on two dataset FB15k-237[28] and WN18RR[8]. FB15k-237 has a larger average node degree than WN18RR, as shown in Table 1. Following [8, 32], we augment the triples each every triple $(h, r, t)$ with *reverse_relation* $(r^-)$ as $(t, r^-, h)$ so we can predict $h$ given $(t, r^-)$.

**Table 1: Dataset**

|  | $|\mathcal{E}|$ | $|\mathcal{R}|$ | $|\mathcal{T}_{\text{training}}|$ | $|\mathcal{T}_{\text{valid}}|$ | $|\mathcal{T}_{\text{test}}|$ | avg degree |
|---|---|---|---|---|---|---|
| WN18RR | 40,943 | 11 | 86,835 | 3034 | 3134 | 3.2 |
| FB15k-237 | 14,541 | 237 | 272,115 | 17,535 | 20,466 | 81 |

## 7.1 Comparing HaSa and HaSa+ to State-of-the-Art KGE models

**Metric:** To evaluate the effectiveness of our method in KGE, we apply it to the link prediction task, which involves predicting the true tails or true head entities in a KG. We use the Mean Rank (MR) of correct entities, Mean Reciprocal Rank (MRR), and Hits at N (Hit@N) which means the proportion of correct entities in top N as metrics to evaluate link prediction results.

**Training process:** 1) We built the embedding function, $f(\cdot)$, using pre-trained LMs with an additional neural network (a linear layer, a Layer normalization, and a dropout layer with probability 0.1 of an element to be zeroed) to reduce the embedding dimension to $d = \{100, 500\}$ for low dimension and high dimension. For the pre-trained LMs, we use BERT-base and sentence-BERT [23](both pre-trained LMs have embedding space 768). BERT-base is commonly used in the knowledge graph embedding [25, 32]. Sentence-BERT [23] has the advantage on sentence prediction and phrase similarity tasks. Our loss function focus on the similarity between query and tail entity so we use sentence-BERT which has a better ability to calculate sentence similarly.

2) We follow the setting of InfoNCE loss [22] to make the aggregation function $g(\cdot)$ as a gated recurrent unit (GRU) neural network.

3) Optimization is done using PyTorch AdamW with learning rate $2 \times 10^{-5}$ and parameter penalty $1 \times 10^{-4}$. The training batch size is $|\mathcal{T}_{batch}| = \{64, 128, 256\}$. The number, $K$, of negative triples per each input triple $(h, r, t)$ depends on the batch size $|\mathcal{T}_{batch}|$. We regard both head entities and tail entities in one batch as negatives excluding the positive tail itself. For each input triple, we also selected the top 3 hardest negatives by computing $\mathbf{e}_{hr}^T \mathbf{e}_t$. Therefore, $K = 5(|\mathcal{T}_{batch}|) - 1$. All experiments are performed on 2 NVIDIA Tesla v100 GPUs and are implemented in Python using the PyTorch framework.

## 7.2 Comparing HaSa with Simple InfoNCE and Hard InfoNCE

We compared our method HaSa (6) performance with the performance of the Simple InfoNCE (2) and Hard InfoNCE (4) on WN18RR and FB15K-237. Table 2 shows the link prediction accuracy after 10 epochs for WN18RR and after 5 epochs for data FB15K-237.

1) Hard InfoNCE can significantly enhance the performance of the Simple InfoNCE in terms of the Hit@1 metric, as it compels

**Table 2: Experimental results on WN18RR and FB15K-237 test set with different contrastive loss functions ($d = 500$, $|\mathcal{T}_{batch}| = 256$}). All methods use the pre-trained LM sentence-BERT as the initialization. Boldface is the best performance and the second-best score is underlined.**

| Loss function | WN18RR | | | | | FB15k-237 | | | | |
|---|---|---|---|---|---|---|---|---|---|---|
| | MR↓ | MRR↑ | Hit@1↑ | Hit@3↑ | Hit@10↑ | MR↓ | MRR↑ | Hit@1↑ | Hit@3↑ | Hit@10↑ |
| $\mathcal{L}_{Simple\_Info}$ (2) | **100** | 0.424 | 0.303 | 0.487 | **0.656** | **151** | 0.277 | 0.186 | 0.306 | 0.466 |
| $\mathcal{L}_{Hard\_Info}$ (4) | 123 | 0.447 | 0.341 | 0.497 | 0.656 | 165 | 0.290 | 0.209 | 0.309 | 0.465 |
| $\mathcal{L}_{HaSa}$ (6) | 123 | **0.452** | **0.351** | **0.501** | 0.650 | 163 | **0.300** | **0.220** | **0.317** | **0.468** |

**Table 3: Experimental results on WN18RR and FB15K-237 test set ($d = 500$, $|\mathcal{T}_{batch}| = 256$}). Boldface is the best performance.**

| Methods | WN18RR | | | | | FB15k-237 | | | | |
|---|---|---|---|---|---|---|---|---|---|---|
| | MR↓ | MRR↑ | Hit@1↑ | Hit@3↑ | Hit@10↑ | MR↓ | MRR↑ | Hit@1↑ | Hit@3↑ | Hit@10↑ |
| Classic KGE | | | | | | | | | | |
| TransE [3] | 2,300 | 0.243 | 0.043 | 0.441 | 0.532 | 323 | 0.279 | 0.198 | 0.376 | 0.441 |
| RotatE [26] | 3,340 | 0.476 | 0.428 | 0.492 | 0.571 | 177 | **0.338** | **0.241** | **0.375** | **0.533** |
| DistMult [36] | 7,000 | 0.444 | 0.412 | 0.470 | 0.504 | 512 | 0.281 | 0.199 | 0.301 | 0.446 |
| ComplexE [29] | 5542 | 0.468 | 0.427 | 0.485 | 0.554 | 546 | 0.278 | 0.194 | 0.297 | 0.450 |
| KBGAN [4] | - | 0.277 | - | - | 0.458 | - | 0.277 | - | - | 0.458 |
| Pre-trained LM based KGE | | | | | | | | | | |
| KGBERT [38] | 97 | 0.216 | 0.041 | 0.302 | 0.524 | 153 | - | - | - | 0.420 |
| StAR [30] | **51** | .401 | 0.243 | 0.491 | 0.709 | **117** | 0.296 | 0.205 | 0.322 | 0.482 |
| LASS [25] | 55 | - | - | - | 0.725 | 131 | - | - | - | 0.479 |
| HaSa (ours) | 135 | 0.489 | 0.400 | 0.535 | 0.666 | 146 | 0.316 | 0.233 | 0.340 | 0.477 |
| HaSa+ (ours) | 112 | **0.538** | **0.444** | **0.588** | **0.713** | 146 | 0.304 | 0.220 | 0.325 | 0.483 |

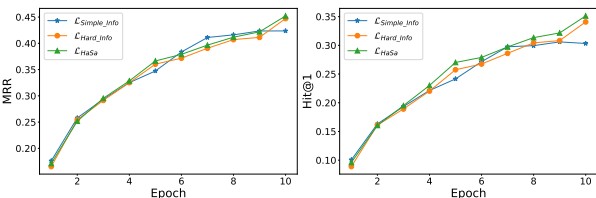

**(a) MRR and Hit@1 for 10 epoch (WN18RR).**

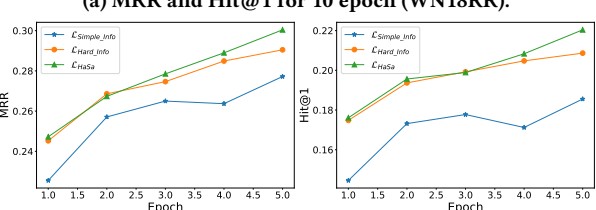

**(b) MRR and Hit@1 for 5 epoch (FB15K-237).**

**Figure 3: Comparing the Simple InfoNCE loss and the Hard InfoNCE loss**

the algorithm to select the negative triple that is easier to classify against the opponent. This observation is consistent with the research that applies contrastive learning with hard negatives in image processing.

2) By considering the false negative triple, HaSa can boost the Hard InfoNCE to improve the performance via various metrics for both data WN18RR and FB15k-237. In particular, concerning the

Hit@1 and MRR metric, HaSa demonstrates significant improvements compared to the Simple InfoNCE.

Figures 3a and 3b show the changes in MRR and Hit@1 scores over the training epochs. For WN18RR, we observe that the performance of Hard InfoNCE and Simple InfoNCE initially appears similar. However. As training progresses, the Hard InfoNCE loss function (4) gradually outperforms the Simple InfoNCE loss function (2).

In the case of FB15K-237, Hard InfoNCE consistently exhibits better performance compared to Simple InfoNCE. Although the HaSa initially shows similar performance to the Hard InfoNCE during the first training epoch, it outperforms Hard InfoNCE as the training process continues.

We compared HaSa and HaSa+ methods to eight other KGE methods: Five classic KGE methods that do not use pre-trained LMs 1) TransE [3], 2) RotateE [26], 3) DisMult [36], 4) ComplexE [29], 5) KBGAN [4], and three KGE methods that use pre-trained LMs 6) KG-BERT[38] with BERT-base, 7) StAR [30] with RoBERTa, 8) LASS [25] with BERT-base. Table 3 shows the results for WN18RR and FB15K-237. We have the following observations:

1) For the WN18RR dataset, HaSa+ achieved the best MRR, H@1, H@3, and H@10 results compared to all KGE methods. Compared to the classic KGE method, the pre-trained LM-based KGE method has a much better result on MR and MRR metrics. However, pre-trained LM-based methods have worse performance on the metric Hit@1. From the significant improvement in metric MR of pre-trained LM-based method, pre-trained LMs provide information to

avoid extremely bad link prediction. However, the trade-off is the ability to perfectly predict the right tails on the metric Hit@1.

2) Compared to the other pre-trained LM-based methods, HaSa and HaSa+ make up for the accuracy on the Hit@1. While hurting the performance on metric MR slightly, it improved the Hit@1 and Hit@3 largely. HaSa+ boosts the HaSa by considering the additional bi-directional loss which makes HaSa see more negative triples. Without considering the bi-directional loss, HaSa also achieves a competitive result. It indeed has the best performance on MRR than other KGE methods.

3) For the FB15K-237 dataset, HaSa and HaSa+ are comparable to state-of-the-art but are generally outperformed by RotateE. However, among the pre-trained LM-based methods, our methods achieve the best result. HaSa has the second-best result on the MRR and Hit@. HaSa+ only boosts the HaSa on metric Hit@10. It does not improve HaSa largely. One possible explanation for the performance difference is that WN18RR and FB15K-237. If we consider the induced graph structure $\mathcal{G}$ of WN18RR and FB15K-237, the average degree of WN18RR is 3.2 while it is 81 for FB15K-237. Therefore, different graph features may be of importance for the two KGs.

## 7.3 Visualizing Embedding Space

We projected the embedding of positive and negative tails into 2-dimensional space using t-SNE, as shown in Figure 4. For WN18RR, we have two queries: (**urban**, **reverse of instance hypernym**) and (**trade**, **member of domain usage**). For FB15k-237, we have two queries: (**rock music**, **music genre artists**) and (**italian**, **reverse of film language**).

Given a query, we observed that the positive tail embeddings are clustered together, and the negative tail embeddings are clustered together. This is consistent with the goal of contrastive learning. For WN18RR, we tend to see two dominant clusters, one for positive tails and one for negative tails. For FB15k-237, we see many smaller clusters for both positive and negative tails scattered in the embedding space. We see from Table 3 that the link prediction result is better on WN18RR dataset than those on FB15k-237. We reason that this is because HaSa is able to learn an embedding space for WN18RR that has more regularity (i.e., less scattered clusters) than FB15k-237

## 7.4 Effect of Hyperparameter $\tau$

As noted previously, for both HaSa and HaSa+, we considered $\tau = p(\ell = \text{fact}|\mathbf{e}_{hr})$ as a hyperparamter. When $\tau = 0$, HaSa is the same as the Hard InfoNCE loss (4). We tested various values of $\tau$: $\{1e-06, 2e-05, 1e-04, 5e-04, 1e-03, 2e-03\}$ and $\{5e-05, 1e-04, 5e-04, 1e-03\}$ for WN18RR and FB15K-237, respectively. For WN18RR, as shown in Figure 5, the model has the best performance on MRR and Hit@10 when $\tau = 2e-05$. For FB15K-237, as shown in Figure 6, the model has a better performance on MRR and Hit@10 when $\tau = 1e-04$.

## 7.5 Effect of Pre-trained LMs

To assess the impact of different pre-trained LMs on HaSa, we employed BERT-base and Sentence-BERT as initialization methods.

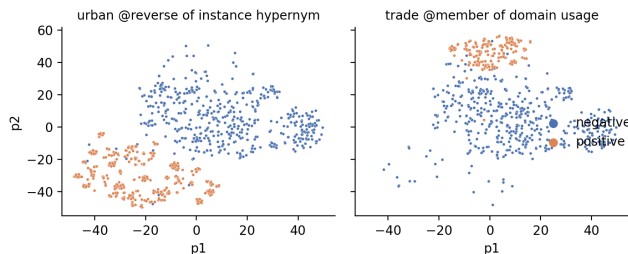

**t-SNE visualization of WN18RR**

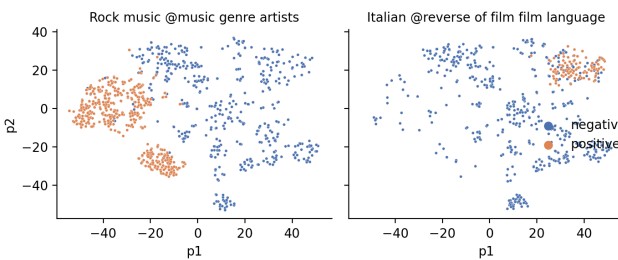

**t-SNE visualization of FB15k-237**

**Figure 4: The 2D visualization of positive and negative tails.**

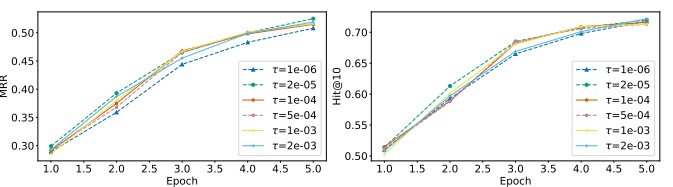

**Figure 5: MRR and Hit@10 for different $\tau$ (WN18RR).**

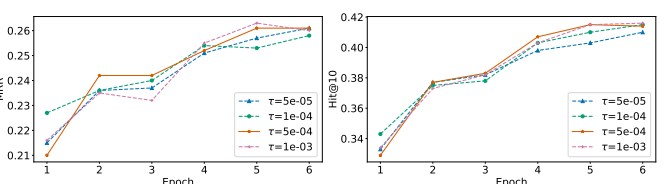

**Figure 6: MRR and Hit@10 for different $\tau$ (FB15K-237).**

Table 4 presents the results for the WN18RR dataset. Utilizing BERT-base yields superior link prediction performance, while Sentence-BERT demonstrates better initialization for training. During the early stages of the training process, Sentence-BERT outperforms BERT-base, as illustrated in Figure 7.

For the FB15K-237 dataset, using Sentence-BERT as the initialization method produces better results compared to the BERT-base, as shown in Table 5. Sentence-BERT's fine-tuning on sentence prediction tasks enhances its performance relative to BERT-base when tasked with calculating similarities between sentences or short phrases

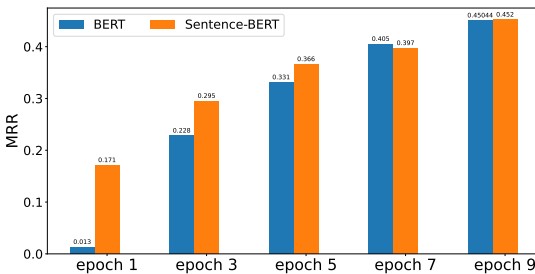

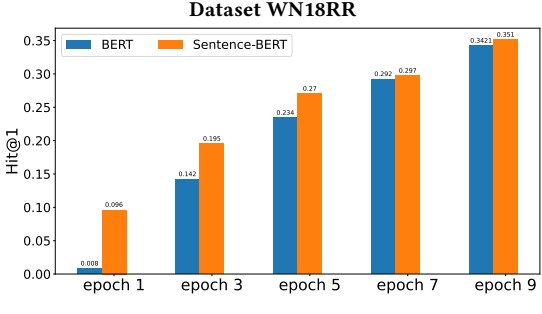

Figure 7: Using the BERT-base and Sentence BERT as the embedding initialization of HaSa

Table 4: Experimental results on WN18RR with different pre-trained LMs.

| Pre-trained LMs | WN18RR | | | |
|---|---|---|---|---|
| | MRR↑ | Hit@1↑ | Hit@3↑ | Hit@10↑ |
| HaSa-Sentence-BERT | 0.452 | 0.351 | 0.501 | 0.650 |
| HaSa-BERT | 0.463 | 0.357 | 0.518 | 0.666 |

Table 5: Experimental results FB15K-237 with different pre-trained LMs.

| Pre-trained LMs | FB15k-237 | | | |
|---|---|---|---|---|
| | MRR↑ | Hit@1↑ | Hit@3↑ | Hit@10↑ |
| HaSa-Sentence-BERT | 0.300 | 0.220 | 0.317 | 0.468 |
| HaSa-BERT | 0.2506 | 0.178 | 0.2757 | 0.383 |

## 7.6 Effect of $d$ and $K$

We considered a low embedding dimension ($d = 100$) and a high embedding dimension ($d = 500$) to compare the link prediction results on the WN18RR dataset. Each dimension had three different batch size settings ($|\mathcal{T}_{batch}| = \{64, 128, 256\}$). We trained each model for only 10 epochs over 7 hours. As shown in Table 6, high-dimensional embeddings consistently outperformed low-dimensional embeddings. We also observed that with more negative triples, high-dimensional embeddings showed significant improvements compared to low-dimensional embeddings.

Note that the number of negative triples depends on the batch size: $K = 5(|\mathcal{T}_{batch}|) - 1$. As shown in Table 6, the larger the batch size, the better the link prediction results. There is a large improvement from batch size 64 to 128. From batch size 128 to 256, the

improvement is also significant but less than the improvement from batch size 64 to 128. This is easy to understand since there are diminishing returns when increasing the number of negatives. Thus, generating higher quality negatives is more important than significantly increasing the number of negative triples.

Table 6: Experimental results on WN18RR with different embedding dimensions and training batch sizes.

| batch size | MR↓ | MRR↑ | Hit@1↑ | Hit@3↑ | Hit@10↑ |
|---|---|---|---|---|---|
| $d=100$ | | | | | |
| 64 | 138 | 0.319 | 0.224 | 0.350 | 0.500 |
| 128 | 158 | 0.378 | 0.281 | 0.411 | 0.570 |
| 256 | 117 | 0.423 | 0.308 | 0.481 | 0.637 |
| $d=500$ | | | | | |
| 64 | 151 | 0.313 | 0.207 | 0.353 | 0.518 |
| 128 | 145 | 0.403 | 0.304 | 0.452 | 0.588 |
| 256 | 123 | 0.452 | 0.351 | 0.501 | 0.650 |

## 8 CONCLUSION AND FUTURE WORK

In this paper, we showed that we might obtain false negative triples when we generate hard negative triples. We noticed that a false negative triple has a smaller shortest path length between the head entity and the tail entity in the knowledge graph. Thus, we proposed the Hardness and Structure-aware (HaSa) contrastive KGE method, which accounts for the false negative triples while generating the hard negative triples. We improved HaSa with HaSa+ by considering the bi-directional loss. Experiments show that they achieve competitive results on WN18RR and FB15K-237 across several metrics.

For future work, we aim to further explore the potential of the bi-directional contrastive loss (the heuristic we used for HaSa+). In terms of false negative tails, we will investigate graph metrics other than the shortest path length. The distribution of false negative tails can also be better approximated with additional information from large language models (LLM). We also plan to study the theoretical aspects of how false negative triplets affect embedding.

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

---

**Algorithm 3:** An algorithm of HaSa+

---

**Input** : Batch of triple $\mathcal{T}_{batch}$, $\tau$, graph structure $\mathcal{G}$, current encoder $f$.

**Output**: The loss $\mathcal{L}_{HaSa+}$.

1 Extract entity batch $\mathcal{E}_{batch}$ from $\mathcal{T}_{batch}$;

2 Extract query set $\{h^-, r^-\}$ from $\mathcal{T}_{batch}$;

**for** $(h, r, t)$ in $\mathcal{T}_{batch}$ **do**

3  $e_h = f(h)$;
  $e_r = f(r)$;
  $e_t = f(t)$;
  $e_{ht} = g(e_h, e_r)$;
  $\mathcal{E}_{hard} = HardNegative(e_{ht}, \mathcal{E}, f)$;
  $\{t_j^-\} = \mathcal{E}_{batch} \cup \mathcal{E}_{hard}/\{t\}$ ;
  $\{s_i^-\}$ = Sampling neighbour nodes based on the distribution $\alpha(\cdot|e_{ht})$ (equation 9);
  Pos = $\exp(e_{ht}^T e_t)$;
  Neg = $\frac{1}{|K|}\sum_{t_j^-} \exp(e_{ht}^T e_{t_j}^-)$;
  FalseNeg = $\frac{1}{|M|}\sum_{s_i^-}\exp(e_{ht}^T e_{s_i}^-)$ ;
  NegHasa = $K(\frac{1}{(1-\tau)}\text{Neg} - \tau\text{FalseNeg})$ ;
  NegHR = $\sum_{h^-, r^-}\exp(e_t^T e_{ht}^-)$;
  Calculate $\mathcal{L}_{HaSa+}(h, r, t)$ for each triple
  $\mathcal{L}_{HaSa+}(h, r, t) =$
  Pos/(Pos + NegHasa) + Pos/(Pos + NegHR) ;

**end**

4 $\mathcal{L}_{HaSa+} = \sum_{(h,r,t)\in\mathcal{T}_{batch}}\mathcal{L}_{HaSa+}(h, r, t)$;

---

# A APPENDIX

## A.1 Remark 1

**Remark 1:** For fix query embedding $e_{hr}$, if we have a false negative tail $e_{t_j}^-$, the gradient of InfoNCE loss w.r.t the $e_{t_j}^-$ will have an opposite direction of positive tails. i.e.

$$\frac{\partial L}{\partial e_{t_j}^-} = e_{hr} - \frac{\partial L}{\partial e_t} \tag{16}$$

PROOF. Considering Simple INfoNCE loss in practice for only one triple,

$$L = -\log\left(\frac{\exp(e_{hr}^T e_t)}{\exp(e_{hr}^T e_t) + \sum_{j=1}^K \exp(e_{hr}^T e_{t_j}^-)}\right). \tag{17}$$

where $e_t$ and $e_{t_j}^-$ are the embeddings of positive tail and negative tail respectively. Derivative the loss function in terms of the embedding,

$$\frac{\partial L}{\partial e_t} = -\frac{\sum_{j=1}^K \exp(e_{t_j}^{-T} e_{hr})e_{hr}}{\exp(e_t^T e_{hr}) + \sum_i \exp(e_{t_j}^{-T} e_{hr})} \tag{18}$$

$$\frac{\partial L}{\partial e_{t_j}^-} = \frac{\exp(t_j^T e_{hr})e_{hr}}{\exp(e_t^T e_{hr}) + \sum_i \exp(e_{t_j}^{-T} e_{hr})} \tag{19}$$

Thus, we have

$$\frac{\partial L}{\partial e_{t_j}^-} - \frac{\partial L}{\partial e_t} = e_{hr} \tag{20}$$

Based on the gradient decent optimizing method, gradient tells us that the positive tail embedding $e_t$ will update by adding a weighted of $e_{hr}$ to minimize the loss. On the contrary, for any negative $e_{t_j}^-$ including the false negative tails, it will be updated by subtracting a weighted $e_{hr}$ to minimize the loss. Thus, contrastive learning attracts similar data together and pushes dissimilar data forward in the embedding space. However, the false negatives $e_{t_j}^-$ will conflict with this goal. □

## A.2 Algorithm of HaSa+

The algorithm for HaSa+ is shown in Algorithm 3.

