# OpenReview forum: "HaSa: Hardness and Structure-Aware Contrastive Knowledge Graph Embedding"
_ACM.org/TheWebConf/2024/Conference — TheWebConf24_

### Official Review · Reviewer_LSGB · 2023-11-03

**Novelty:** 5
**Technical Quality:** 4

**Review:**

The paper introduces a novel method of mining true hard negative samples used in contrastive learning in the area of knowledge graph embedding (KGE). This method of estimating the "true" negatives is heuristic-based, which looks at the correlation between the distribution of the number of hops and the number of false negatives in the benchmark datasets used in this work, WN18RR and FB15k-237. The novelty lies in incorporating the heuristic, that the distance from head to tail entity (i.e., number of hops) has a negative correlation to the number of false negatives, into the optimization step with sense-making techniques such as important sampling and the law of total expectation.

Typo:
1. line 672: However. As training progresses, ... -> However, as training progresses, ...

**Questions:**

While the evaluation shows improvement over other methods in the link prediction task, a couple of questions that have an impact on the interpretability of the results:
1. In line 331 it is mentioned that identifying true/false negatives by splitting the dataset into a "retain" and a "missing" partition is a weak assumption, of which the whole methodology of reducing false negatives is built on top (i.e., negative correlation between head-tail entity
distance and number of false negatives). How reliable is this estimation in terms of generalizability? Does this heuristic also apply to other knowledge graphs with a more complex graph structure? Specifically, the negative correlation mentioned above is found within the two datasets tested in this work, and incorporating this observation naturally creates better feature representation because equation 9 scales down the impact of having false negatives in contrastive learning loss.
2. Building on top of the previous question, in line 130-131, the authors claim that HaSa and HaSa+ have comparable results to other SOTA for KGE. However, I am not sure whether this is too big of a claim to improve KGE overall with the proposed methodology, which is
based on observation from the tested benchmarks. If the authors want to make a general claim about KGE, a descriptive and more in-depth analysis (e.g., how imbalanced the use relations is; what kind of graph structure the graphs depict...) on the graphs used in this
work is needed, as well as evaluation of other tasks such as entity linking.
3. In line 430-431, it is claimed that the shortest path length can distinguish between true and false negative triples. Based on the first question, is an observation on a dataset which is benchmarked & which is based on a weak assumption, robust enough to make such a
claim?
4. In line 505-506, the authors mentioned that reference [5] (Ting Chen, Simon Kornblith, Mohammad Norouzi, and Geoffrey Hinton. 2020. 919 A simple framework for contrastive learning of visual representations. In Inter- 920 national conference on machine learning. PMLR, 1597–1607.) has used a bi-directional contrasting method. Can the authors elaborate more on this? In my understanding in contrastive learning literature, reference 5 states that augmentation via some specific image transformation techniques are more important than the other, and stacking up two different transformations could help. Where is the "bi-directional contrasting" mentioned?

**Ethics Review Description:**

-

**Reviewer Confidence:**

2: The reviewer is willing to defend the evaluation, but it is likely that the reviewer did not understand parts of the paper

**Scope:**

4: The work is relevant to the Web and to the track, and is of broad interest to the community

---

### Official Review · Reviewer_quLS · 2023-11-12

**Novelty:** 4
**Technical Quality:** 4

**Review:**

Summary:
This paper focuses on the bias from generating negative triples in KGE, particularly false negative triples. The paper suggests that creating high-quality negative triples may inadvertently increase the number of false negatives. To counter this, the authors introduce the Hardness and Structure-aware (HaSa) method to minimize the impact of false negatives while generating challenging negative triples. Experiments demonstrate that HaSa achieves top results on the WN18RR and FB15k-237 datasets compared to traditional and pre-trained language model-based KGE methods.

Weakness:
1. The performance on the FB15k-237 dataset is not good. The SOTA performance is about 0.42, so I think the performance on FB15k-237 dataset is not convincing enough to demonstrate the superiority of HaSa.
2. The presentation/writing needs significant improvement. The related work section is too short and there’s no method overview figure to help readers understand the methodology.
3. The baseline methods are outdated, the most recent classical KGE baseline was proposed in 2017.
4. Missing important pre-trained LM based KGE method like SimKGC [1], kgt5 [2], KEPLER [3]
[1] Wang, Liang, et al. "SimKGC: Simple Contrastive Knowledge Graph Completion with Pre-trained Language Models." Proceedings of the 60th Annual Meeting of the Association for Computational Linguistics (Volume 1: Long Papers). 2022.
[2] Saxena, Apoorv, Adrian Kochsiek, and Rainer Gemulla. "Sequence-to-Sequence Knowledge Graph Completion and Question Answering." Proceedings of the 60th Annual Meeting of the Association for Computational Linguistics (Volume 1: Long Papers). 2022.
[3] Wang, Xiaozhi, et al. "KEPLER: A unified model for knowledge embedding and pre-trained language representation." Transactions of the Association for Computational Linguistics 9 (2021): 176-194.
5. Can the proposed HaSa be applied to classical KGE methods? It would be interesting to see that performance to demonstrate the versatility of HaSa.
6. In figure 3, compared to simple_info and hard_info, HaSa’s performance gain seems limited. I would encourage the authors to discuss the mechanism of this phenomenon.
7. Why did the authors present experiment 7.4 in the paper? This section just select an optimal hyperparameter without presenting any insight. I can’t see the purpose of this section.

**Questions:**

N/A

**Reviewer Confidence:**

3: The reviewer is confident but not certain that the evaluation is correct

**Scope:**

3: The work is somewhat relevant to the Web and to the track, and is of narrow interest to a sub-community

---

### Official Review · Reviewer_h4dR · 2023-11-13

**Novelty:** 4
**Technical Quality:** 4

**Review:**

Summary:
- This paper propose a hardness and structure-aware contrastive KGE method for Pre-trained LM based KGE. HaSa alleviate the effect of false negative triples while generating the hard negative triples with mathematical derivation and code implementation.

Strengths:
- Negative sampling is an important and interesting topic in the KGE research
- The author's derivation is theoretical and the implementation is reasonable

Weaknesses:
- Some important baselines are missing in the experiments. For the classic KGE methods, the baselines are relatively early and many strong baselines in the recent years are not included in the table. For example, PairRE[1], DualE[2]
- For the PLM-baes KGE methods, some baselines are still missing. For example, SimKGC[3] also employ constrastive learning and new negative sampling methods for PLM-based KGC. But its results are not reported in the table.
- In the baseline methods, the negative sampling methods are missing. For example, NSCaching [4], SANS [5]. As a new negative sampling metho, HaSa should make comprehensive comparasion with other negative sampling and constrastive learning based method
- The exepriments employ SentenceBERT as backbone but the baselines are bert-based methods. Different Backbones may have performance implications, and I would suggest using the same backbone for the different PLM-based KGC methods of backbones to highlight HaSa's effects.
- There are some typos in the paper. For example, ComplexE --> ComplEx


[1] PairRE: Knowledge Graph Embeddings via Paired Relation Vectors
[2] Dual Quaternion Knowledge Graph Embeddings
[3] SimKGC: Simple Contrastive Knowledge Graph Completion with Pre-trained Language Models
[4] NSCaching: simple and efficient negative sampling for knowledge graph embedding
[5] Structure aware negative sampling in knowledge graphs

**Questions:**

Please address the issues mentioned in my review. If all my concerns are resolved, I will consider raising my score.

**Reviewer Confidence:**

4: The reviewer is certain that the evaluation is correct and very familiar with the relevant literature

**Scope:**

4: The work is relevant to the Web and to the track, and is of broad interest to the community

---

### Official Review · Reviewer_evs2 · 2023-11-22

**Novelty:** 6
**Technical Quality:** 3

**Review:**

In this paper, the authors propose a new method for knowledge graph completion and evaluate its performance experimentally. The characteristics of false negative triples reported in this paper are exciting and worth reporting as new findings. On the other hand, the experimental results of the proposed method based on the findings do not clearly show any improvement over the conventional methods.

Pros:
New findings of the characteristics of the false negative triple are reported.
Various experiments have provided detailed evaluations.

Cons:
It does not outperform conventional methods in terms of performance.
Lacks fair experimental evaluation and description

The authors claim that "archives state-of-the-art results for several metrics for WN18RR" (abstract) and show that the results for HITS@10 are in bold in Table 3. However, they are lower than those of LASS. Although the results for the other metrics are not reported in the LASS paper, it is quite possible that the performance of the proposed method does not exceed that of LASS. The source code of LASS is publicly available, so experimental results for other metrics such as Hit@1 should be reported if the authors claim "archives state-of-the-art results". Also, StAR results are better when the ensemble model is used as reported in the paper.

Minor comments:
There is a missing discussion of studies on Negative Sampling on KG such as followings.
https://doi.org/10.1109/ICDE.2019.00061
https://doi.org/10.48550/arXiv.2202.09606
https://doi.org/10.1007/s10844-023-00796-y
https://doi.org/10.1016/j.eswa.2022.117361

**Questions:**

Line 709 “However, … our method achieves the best result”
Since it is stated without pointing to a specific result, it could be taken as a summary statement of the overall result, but on what basis is it making this assertion?

Thank you very much for your comments.

**Reviewer Confidence:**

3: The reviewer is confident but not certain that the evaluation is correct

**Scope:**

3: The work is somewhat relevant to the Web and to the track, and is of narrow interest to a sub-community

---

### Official Review · Reviewer_NdCY · 2023-11-25

**Novelty:** 5
**Technical Quality:** 3

**Review:**

This paper initiates an investigation into reducing false negative triples in generating hard negative triples. Moreover, the proposed method improves the effectiveness of InfoNCE for KGE. Additionally, this paper offers a clear and coherent narrative, featuring a well-structured framework that is substantiated by both data and theoretical foundations. It effectively explains the methods and motivations employed in the study.

Prods:
1. This paper first explores how to decrease false negative triples while generating hard negative triples, contributing to better performance of InfoNCE for KGE.

2. The paper presents a lucid narrative. It has a well-organized structure supported by data and theoretical foundations, effectively elucidating the methods and motivations.

Cons:
1. The designed method is relatively simple, and the improvement in effectiveness is limited.

2. The experiments are not comprehensive enough.
(a) There is a limited comparison with baselines, for instance, SimKGC (https://arxiv.org/pdf/2203.02167) is not included in the comparison.
(b) Experiments on large-scale datasets, such as Wikidata5M, are lacking.
(c) There is no exploration of HaSa’s performance on structure-based models.

**Questions:**

Q1. Whether HaSa could still improve the performance of InfoNCE on large-scale datasets?

Q2. Could HaSa be used on structure-based methods?

**Reviewer Confidence:**

3: The reviewer is confident but not certain that the evaluation is correct

**Scope:**

4: The work is relevant to the Web and to the track, and is of broad interest to the community

---

### Decision · Program_Chairs · 2024-01-22

**Decision:**

Accept

**Comment:**

This paper introduces a method for Pre-trained LM-based KGE, termed "Hardness and Structure-aware Contrastive Knowledge Graph Embeddings (KGE)." It explores mitigating false negative triples in the generation of hard negative triples. The study suggests that generating high-quality negative triples could inadvertently elevate the count of false negatives. To address this concern, the authors propose the Hardness and Structure-aware (HaSa) method, aiming to minimize the influence of false negatives while creating challenging negative triples.

 The authors provide further experimental results to address corresponding criticism of the reviewers.

 Pros:
 1. The paper is well written, well structured, and easy to read.
 2. Negative sampling is an important and interesting topic in the KGE research
 3. The paper features a well-structured framework that is substantiated by both data and theoretical foundations.
 4. New findings of the characteristics of the false negative triple are reported. This paper first explores how to decrease false negative triples while generating hard negative triples, contributing to better performance of InfoNCE for KGE.
 5. Various experiments have provided detailed evaluations.


 Cons:
 1. The designed method is relatively simple
 2. The proposed approach does not outperform conventional methods in terms of performance. The improvement in effectiveness is limited.
 3. The reviewers have pointed out some deficiencies in the experimental evaluation, which has been addressed by the authors